# The effect of molnupiravir and nirmatrelvir on SARS-CoV-2 genome diversity in severe models of COVID-19

Rebekah Penrice-Randal,[1] Eleanor G. Bentley,[1] Parul Sharma,[1] Adam Kirby,[1] I'ah Donovan-Banfield,[1,2] Anja Kipar,[1,3] Daniele F. Mega,[1] Chloe Bramwell,[1,4] Joanne Sharp,[4,5] Andrew Owen,[4,5] Julian A. Hiscox,[1,2,6] James P. Stewart[1,5]

**ABSTRACT**  Immunocompromised individuals are susceptible to severe coronavirus disease 2019 and potentially contribute to the emergence of variants with altered pathogenicity due to persistent infection. This study investigated the impact of immunosuppression on severe acute respiratory syndrome coronavirus 2 (SARS-CoV-2) infection in K18-hACE2 mice and the effectiveness of antiviral treatments in this context during the first 7 days of infection. Mice were immunosuppressed using cyclophosphamide and infected with a B lineage of SARS-CoV-2. Molnupiravir and nirmatrelvir, alone and in combination, were administered, and viral load and viral sequence diversity were assessed. Treatment of infected but immunocompromised mice with both compounds either singly or in combination resulted in decreased viral loads and pathological changes compared to untreated animals. Treatment also abrogated infection of neuronal tissue. However, no consistent changes in the viral consensus sequence were observed, except for the emergence of the S:H655Y mutation. Molnupiravir, but not nirmatrelvir or immunosuppression alone, increased the transition/transversion ratio, representative of G > A and C > U mutations, and this increase was not altered by the co-administration of nirmatrelvir with molnupiravir. Notably, immunosuppression itself did not appear to promote the emergence of mutational characteristics of variants of concern (VOCs). Further investigations are warranted to fully understand the role of immunocompromised individuals in VOC development, especially by taking persistence into consideration, and to inform optimized public health strategies. It is more likely that immunodeficiency promotes viral persistence but does not necessarily lead to substantial consensus-level changes in the absence of antiviral selection pressure. Consistent with mechanisms of action, molnupiravir showed a stronger mutagenic effect than nirmatrelvir in this model.

**IMPORTANCE**  The central aim of this study was to risk-assess the impact of administering a mutagenic antiviral compound, molnupiravir, to patients believed to already be at risk of generating increased viral diversity, which could have severe implications for antiviral resistance development. Combination therapy has a long history of mitigating antiviral resistance risk and was used in this study to demonstrate its potential usefulness in a severe acute respiratory syndrome coronavirus 2 (SARS-CoV-2) context. Animals treated with molnupiravir showed an increase in transition/transversion ratios over time, consistent with the drug's mechanism of action and a recent UK-wide phase II clinical trial assessing the efficacy of molnupiravir in humans. The addition of nirmatrelvir increased viral clearance, which in turn reduces the probability of viral persistence and rapid intra-host evolution of SARS-CoV-2.

**KEYWORDS**  SARS-CoV-2, COVID-19, immunocompromised, intra-host evolution, molnupiravir, nirmatrelvir, Paxlovid

**Peer Reviewer** Kaan Çeylan, Faculty of Medicine University of Gaziantep, Gaziantep, Turkey

Address correspondence to Rebekah Penrice-Randal, rebee@liverpool.ac.uk.

Rebekah Penrice-Randal and Eleanor G. Bentley contributed equally to this article. Author order was determined randomly.

A.O. is a director of Tandem Nano Ltd and co-inventor of patents relating to drug delivery. A.O. has been co-investigator on funding received by the University of Liverpool from ViiV Healthcare and Gilead Sciences in the past 3 years unrelated to COVID-19. A.O. has received personal fees from Gilead and Assembly Biosciences in the past 3 years, also unrelated to COVID-19. J.P.S. has received funding from ENA Respiratory Pty Ltd, Bicycle Tx Ltd, and Infex Therapeutics Ltd unrelated to this study. R.P.-R. is an employee at TopMD Precision Medicine Ltd. No other conflicts are declared by the other authors.

See the funding table on p. 14.

Unsurprisingly, since the start of the severe acute respiratory syndrome coronavirus 2 (SARS-CoV-2) pandemic and the first deposited genome sequences, and like other coronaviruses, SARS-CoV-2 has diverged through single nucleotide polymorphism and homologous and heterologous recombination applications resulting in insertions and deletions (1, 2). Over the course of the pandemic, changes that have dominated have resulted in increased transmissibility, such as the P323L/D614G changes in early 2020 (3–5), immune evasion (6), and altered pathogenicity (7).

Founder effects, population bottlenecks, selection pressures, and behavior have contributed to the diversification of the SARS-CoV-2 genome but also to the apparent waves of different variants. Several variants of concern (VOCs) have arisen that have a transmission advantage and/or potential immune evasion. Some reports have suggested that such variants may have arisen in hosts with compromised immunity and/or persistent infections, where infection leads to the generation of more diverse variants through longer viral evolution within an individual (8). This includes a changing landscape of dominant viral genome sequence and minor genomic variants in immunocompromised individuals, e.g., in a patient with cancer (9). Changes within the individual mapped to several different regions on the SARS-CoV-2 genome, including the spike glycoprotein and orf8.

Complicating the picture of potential rapid and dramatic genomic change in immunocompromised hosts is that similar changes can be observed in immunocompetent patients. This can be either as part of the dominant genomic sequence (10) or minor variant genomes (1). Indeed, genomic variants with deletions can be identified in the minor genomic variant population of Middle East respiratory syndrome coronavirus from patients (11) and as part of the dominant genomic sequence in camels (12, 13).

Parallels with other animal coronaviruses can be found where persistent infections are established, and this might be associated with pathogenicity; examples are feline coronavirus infections and feline infectious peritonitis (14–17). Thus, one concern with long-term persistence of SARS-CoV-2 in immunocompromised patients is that new transmissible variants could emerge (8).

Three small molecule direct-acting antivirals have received early use authorization for the treatment of coronavirus disease 2019 (COVID-19): remdesivir, molnupiravir (both nucleoside analogs which target viral nucleic acid synthesis), and nirmatrelvir (which targets the main viral protease). Unlike remdesivir, molnupiravir and nirmatrelvir are orally administered and thus more readily deployed for treatment in the community. Nirmatrelvir is packaged with ritonavir (as Paxlovid), the later molecule acts as a pharmacokinetic boosting agent to inhibit P450 (CYP) 3A4. However, adequate nirmatrelvir plasma concentrations can be achieved in mice without the need for ritonavir boosting. In cell culture, single or combination treatment can result in decreased viral replication (18, 19), and a natural extension is that such antivirals may be deployed as combination therapy to reduce the emergence of resistant genotypes (20). Resistant genotypes/phenotypes have been identified *in vitro* for remdesivir (21). Molnupiravir has previously been shown to enhance viral transition/transversion (Ts/Tv) mutations in a phase II clinical trial (22), and a molnupiravir-associated signature has been identified in circulating SARS-CoV-2 lineages since the introduction of molnupiravir in 2022 (23).

Immunocompromised patients with a SARS-CoV-2 infection are treated as a priority with antivirals, including those compounds that generically target virus replication by causing hyper-mutation or specifically preventing the function of a viral protein critical to the life cycle of the virus. Such antivirals may be deployed as combination therapy to reduce the emergence of resistant genotypes (20) and may be particularly relevant for patients with compromised immunity (24). However, in the latter patients, antivirals may decrease viral loads but enhance genomic plasticity. To investigate this, the genomic variation of SARS-CoV-2 was evaluated in an immunocompromised host over the first 7 days of infection, in the absence and presence of medical countermeasures. We have developed animal models of COVID-19 to be able to assess pathogenicity of new variants

and develop interventions (25–27). An immunosuppressed K18-hACE2 transgenic mouse model was used to simulate patients with severe COVID-19 (28, 29). Two antivirals, molnupiravir and nirmatrelvir, were evaluated either singly or in combination.

## MATERIALS AND METHODS

### Animal infection and treatment

A UK variant of SARS-CoV-2 (hCoV-2/human/Liverpool/REMRQ0001/2020) was used as described previously (30, 31). Mutations belonging to the B lineage virus are outlined in Table 1.

Transgenic mice carrying the human ACE2 gene under the control of the keratin 18 promoter (K18-hACE2; formally B6.Cg-Tg(K18-ACE2)2Prlmn/J) were purchased from Jackson Laboratories (France) at 8–10 weeks of age. Mice were maintained under specific-pathogen-free (SPF) barrier conditions in individually ventilated cages and underwent a week of acclimatization in these conditions prior to experimental use.

Experimental design is shown in Fig. 1, and treatment groups are detailed in Table 2. Animals were randomly assigned into multiple cohorts of four animals using a random number generator. For operational reasons, at high containment, the treatment groups were not blinded during the experiment. The sample size was determined using prior experience of similar experiments with SARS-CoV-2. For SARS-CoV-2 infection, mice were anesthetized lightly with isoflurane and inoculated intranasally with 50 µL containing $10^4$ PFU SARS-CoV-2 in phosphate buffered saline (PBS) as described previously (26). Some cohorts of mice were immunosuppressed by treatment with cyclophosphamide (100 mg/kg) intra-peritoneally at days −4 and −1 pre-infection. Molnupiravir was made up in 10% PEG400 and 2.5% cremophor in water and used at 100 mg/kg. Nirmatrelvir was dissolved in 2% Tween 80 in 98% (vol/vol) of 0.5% methyl cellulose and used at 500 mg/kg. These doses were chosen based on the known therapeutic range for these drugs in mice (32–35). Both drugs were administered via the oral route 1 h prior to infection and then twice daily up to 4 days post-infection via the oral route. Groups of animals were kept in the same cages during the experiment and were always weighed and treated in the same order. Mice were sacrificed at day 6 (vehicle- and cyclophosphamide-treated group) or 7 (all others) after infection by an overdose of pentobarbital. Weights were recorded daily, and tissues were removed immediately for downstream processing. The right lung and nasal turbinates were frozen at −80°C until further processing. The left lung and heads were fixed in 10% neutral buffered formalin for 24–48 h and then stored in 70%. No data were excluded from the analyses.

### Histology, immunohistology, and morphometric analysis

The fixed left lung was routinely paraffin wax embedded. Heads were sawn longitudinally in the midline using a diamond saw (Exakt 300; Exakt), and the brain was left in the skull. Heads were gently decalcified in RDF (Biosystems) for twice 5 days, at room temperature and on a shaker, then both halves paraffin wax embedded. Consecutive sections (3 µm–5 µm) were either stained with hematoxylin and eosin or used for immunohistology (IH). IH was performed to detect viral antigen expression using the

TABLE 1  Input virus used in this study

| | Input virus | |
| --- | --- | --- |
| Nucleotide change | Gene | Amino acid change |
| A6948C | Nsp3 | N1410T or N2228T |
| G11083T | Nsp6 | L37F or L3606F |
| C21005T | Nsp16 | A116V or A2513V |
| C25452T | Orf3a | I20 no change |
| C28253T | Orf8 | F120 no change |

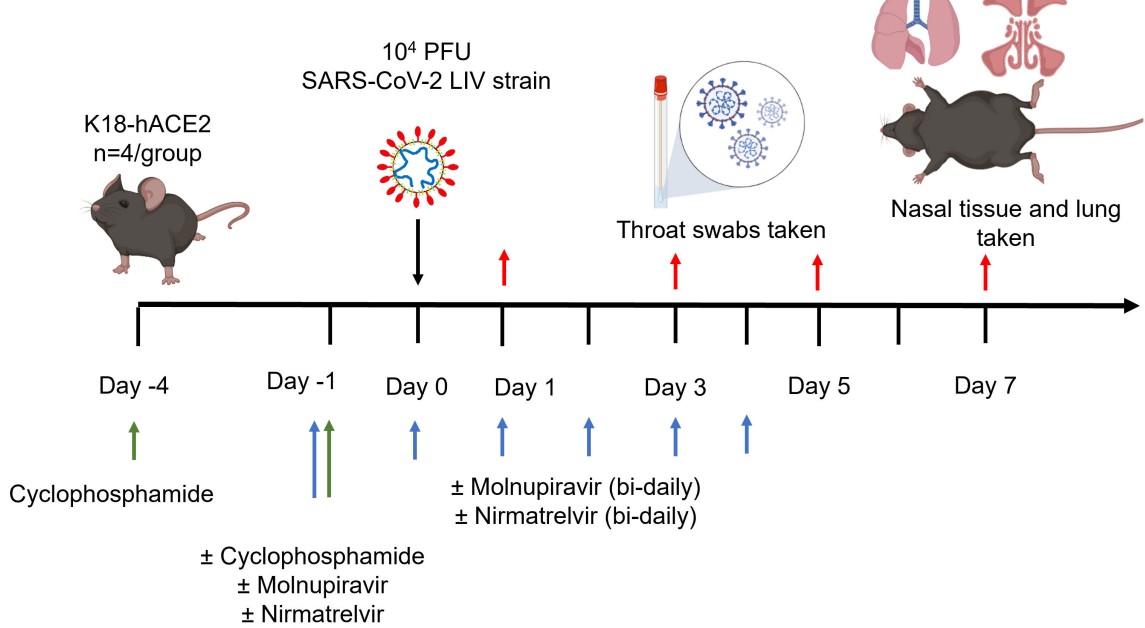

**FIG 1** Schematic diagram of the experimental design for infection of immunocompromised K18-hACE2 mice with SARS-CoV-2 and evaluation of two antiviral drugs given at a human equivalent dose; molnupiravir, a broad-acting compound causing error catastrophe, or nirmatrelvir, which specifically targets the viral 3C-like protease. Cyclophosphamide was used at 100 mg/kg via the intraperitoneal route to immunosuppress mice. Molnupiravir was used at 100 mg/kg and nirmatrelvir was used at 500 mg/kg both via the oral route. Effects of infection and treatment were evaluated by measuring the weight of the mice daily, determining viral loads in sequential oral/throat swabs and at day 7 post-infection, and examining nose, brain, and lung at day 7 post-infection for any histological changes and the expression of SARS-CoV-2 nucleoprotein.

horseradish peroxidase method and a rabbit anti-SARS-CoV nucleocapsid protein (Rockland, 200-402-A50) as primary antibody, as previously described (26, 36, 37).

For morphometric analysis, the immunostained sections were scanned (NanoZoomer-XR C12000; Hamamatsu, Hamamatsu City, Japan) and analyzed using the software program Visiopharm (Visiopharm 2020.08.1.8403; Visiopharm, Hoersholm, Denmark) to quantify the area of viral antigen expression in relation to the total area (area occupied by lung parenchyma) in the sections. This was used to compare the extent of viral antigen expression in the lungs between the different treatment groups. A first app was applied that outlined the entire lung tissue as region of interest (ROI; total area). For this, a Decision Forest method was used, and the software was trained to detect the lung tissue section (total area). Once the lung section was outlined as ROI, the lumen of large bronchi and vessels was manually excluded from the ROI. Subsequently, a second app with the Decision Forest method was trained to detect viral antigen expression (as brown 3,3′-diaminobenzidin, i.e. DAB precipitate) within the ROI.

**TABLE 2** Treatment groups for *in vivo* analysis

| Group | Treatment |
| --- | --- |
| 1 | Control (vehicle) |
| 2 | Cyclophosphamide |
| 3 | Molnupiravir |
| 4 | Cyclophosphamide + molnupiravir |
| 5 | Cyclophosphamide + nirmatrelvir |
| 6 | Cyclophosphamide + molnupiravir + nirmatrelvir |

## RNA extraction

To inactivate the virus in throat swabs, 260 µL of swab buffer was inactivated in a Class II Biosafety cabinet using 750 µL of TRIzol LS reagent (ThermoFisher, Runcorn, UK), and transferred into 2 mL screw-cap vials and mixed. Samples were stored at −80°C until further analysis. RNA samples were normalized to 20 ng/µL before qPCR and sequencing.

## qRT-PCR for viral load

Viral loads were quantified using the GoTaq Probe 1-Step RT-qPCR System (Promega). For quantification of SARS-CoV-2 RNA, the nCOV_N1 primer/probe mix from the SARS-CoV-2 (2019-nCoV) CDC qPCR Probe Assay (IDT) was utilized alongside murine 18S primers, as described previously (25, 26).

## Sequencing of SARS-CoV-2

RNA samples were placed into plates based on very high viral load (Ct < 18), high viral load (Ct 20–24), medium viral load (Ct 25–28), or low viral load (Ct > 28) to assist with pooling strategy. Library preparation consisted of converting RNA to cDNA using LunaScript (ThermoFisher), then amplified by reverse complement-PCR amplification (EasySeq SARS-CoV-2 Whole Genome Sequencing kit, Nimagen, Netherlands). This kit barcodes and ligates Illumina adapters in a single PCR reaction, with two separate pools of primers (pools 1 and 2). After amplification, each amplicon library was pooled 1:1 before being cleaned with AmpliClean beads and quantification. The two pools were then added together and denatured, where 2 µL, 4 µL, 8 µL, and 16 µL of each pool for very high, high, medium, and low viral load were taken, respectively. Finally, the denatured amplicon library was loaded into the NovaSeq cartridge (2 × 150 bp run).

## Bioinformatics

Figure S1 provides an overview of the workflow used in this study. In short, raw paired-end fastq files were inputted into the EasySeq pipeline (version 0.9, code available at: https://github.com/JordyCoolen/easyseq_covid19), which performs quality control steps, maps to the reference genome (Wuhan-Hu-1; NC045512.2), calls variants, and generates a consensus genome for each sample (38). Default parameters were used and are as follows: variant call threshold = 0.5; variant calling quality threshold = 20; variant calling minimum depth = 10. Pangolin (version 4.0.6) was used to assign SARS-CoV-2 lineage, with maximum ambiguity set at 0.3 (38, 39). Consensus sequences were also inputted into Nextclade (40) for lineage assignment and primer-trimmed bam files and their associated index file were inputted into DiversiTools (https://github.com/joseph-hughes/DiversiTools) to assess global minor variation. Sequencing data were analyzed as previously described, and statistical analysis and visualization were performed in R (22). DiversiTools facilitates an in-depth analysis of viral diversity in each sample, as opposed to only the consensus/dominant genomic information, as previously described (5, 22, 41, 42). The minor variant analysis was conducted on samples where the dominant genome sequence had a minimum 90% consensus called and 90% of genome positions had a minimum coverage of 100×. Entropy outputs from DiversiTools were imported to R, and for each genome, the mean transition and transversion were calculated, followed by the mean transition/transversion ratio to assess global minor variants. Similarly, the average for each minor base change across the genome was calculated to identify biases in base changes associated with molnupiravir and nirmatrelvir treatment. Raw fastq files are available under SRA Project Accession: PRJNA886870. Code for analysis and figure generation is available at https://github.com/Hiscox-lab/viral-genomics-immunosupression-and-countermeasures.

## Statistics

Graphs were prepared, and statistical analyses were performed using Prism 10 (GraphPad Inc.). *P*-values were set at 95% CI. A repeated-measures two-way analysis of variance

(ANOVA) (Bonferroni post-test) was used for time courses of weight loss; log-rank (Mantel-Cox) test was used for survival curve and Mann-Whitney U-test for side-by-side comparisons. All differences not specifically stated to be significant were not significant ($P > 0.05$). For all figures, $*P < 0.05$.

## RESULTS AND DISCUSSION

Since the emergence of the Alpha VOC, there has been discussion on the involvement of the immunocompromised host and the generation of variants (8, 43–47). There are many case studies in the literature that follow SARS-CoV-2 evolution in immunocompromised hosts; however, the generation of VOCs is likely due to persistent infection as opposed to the immunocompromised immune system itself. Little has been explored experimentally. In this study, mice were chemically immunocompromised with cyclophosphamide, which is known to efficiently remove adaptive immunity in the form of B and T cells (48). Additionally, therapeutic agents, molnupiravir and nirmatrelvir, were used independently and in combination to determine the effectiveness of these compounds in an immunocompromised model and the impact of these compounds on viral sequence diversity during the first 7 days of infection.

Modeling an immunocompromised state in animal models in the context of SARS-CoV-2 is important for the consideration of countermeasures that may be utilized for humans who are considered vulnerable. Cyclophosphamide has been used previously to study the impact of immunosuppression in a hamster model (49–51), where intranasally infected hamsters with cyclophosphamide treatment before infection had prolonged weight loss and an inadequate neutralizing antibody response to SARS-CoV-2. Distinct transcriptional profiles were identified between immunocompetent and immunosuppressed animals; however, the impact of antivirals or viral genome diversity was not investigated.

To investigate the frequency of genomic changes that occur in SARS-CoV-2 in the immunocompromised or competent host in the presence or absence of antiviral drugs, K18-hACE2 transgenic mice were used as a model for severe SARS-CoV-2 infection in humans (52). We have found that the pathological changes in the lungs in this model, in many aspects, resemble those in humans who have died of severe COVID-19 (26, 28, 29, 36, 37). To mimic a host with compromised immunity, an experimental protocol was developed in which mice were exposed to cyclophosphamide (48) (Fig. 1; Table 2). Several antiviral regimens in humans were simulated in the mouse model by giving a human equivalent dose of either molnupiravir (100 mg/kg), nirmatrelvir (500 mg/kg), or both in combination. This included prophylactic followed by therapeutic treatment. Mice were infected with $10^4$ PFU of SARS-CoV-2.

## Treatment with Molnupiravir or Nirmatrelvir either individually or in combination provides recovery in immunocompromised mice infected with SARS-CoV-2

Cyclophosphamide treatment prior to SARS-CoV-2 infection of hACE2 mice led to a more pronounced early weight loss in comparison to immunocompetent mice, a phenomenon previously reported in hamsters (51). This was not associated with earlier mortality than in vehicle-treated immunocompetent mice, although in humans, a delayed adaptive immune response has been shown to be associated with fatality in COVID-19 patients, which may have been observed over longer time frames (53). Daily weighing of the animals indicated that all groups lost body weight after day 1 (Fig. 2). We attribute this to aversion to eating, as all therapies were applied by gavage. However, starting at day 3, all groups, except for mice exposed to cyclophosphamide or mice exposed to cyclophosphamide and treated with molnupiravir, started to gain or stabilize weight. By days 5 and 6, a clear pattern had emerged where all groups treated with molnupiravir or nirmatrelvir either individually or in combination had regained their starting weight. The exception to this were mice exposed to vehicle only (controls) or cyclophosphamide; these reached a humane end point on day 6 (Fig. 2). Comparison of survival curves again

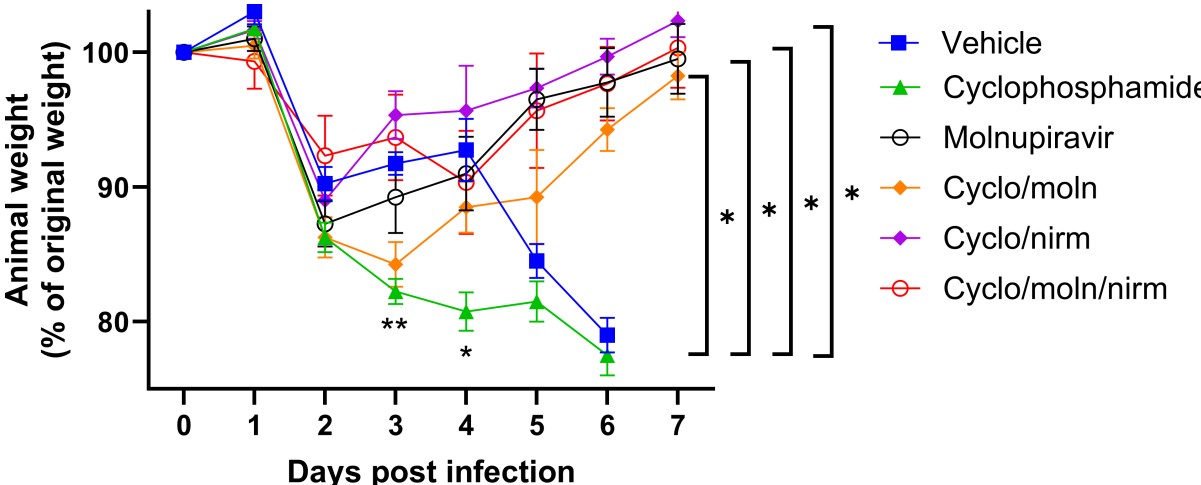

**FIG 2** Treatment of SARS-CoV-2-infected mice leads to decreased weight loss. K18-hACE2 mice were challenged intranasally with $10^4$ PFU SARS-CoV-2. Mice were monitored for weight at indicated time points. ($n = 4$). Data represent the mean residual weight ± SEM. Comparisons were made using a repeated-measures two-way ANOVA (Bonferroni post-test). Asterisks below the curves represent *$P < 0.05$ and **$P < 0.01$ between the cyclophosphamide and vehicle groups. Brackets and asterisks at the side represent $P < 0.05$ for the vehicle/cycophosphamide groups and the drug-treated groups.

indicated that immunocompromised animals treated either singly or in combination with each therapeutic went on to survive (Fig. 3).

## Viral load decreases in immunocompromised mice treated with Molnupiravir or Nirmatrelvir either individually or in combination

Viral load in terms of copy numbers of the SARS-CoV-2 genome were calculated for throat swabs during infection and compared to nasal tissue and lung tissue at the end of the experiment. The data indicated that for throat swabs on days 1 and 3 post-infection, there was a significant decrease in viral load in animals treated with molnupiravir or nirmatrelvir either individually or in combination compared to untreated controls (Fig. 4A). On day 3, there was a significant difference between the combination of both compounds and molnupiravir alone (Fig. 4A). No significant differences were observed between vehicle control and cyclophosphamide-only groups.

Comparison of viral loads and titers in nasal and lung tissue, respectively (Fig. 4B and C, respectively), at day 7 post-infection reflected that there was a significantly lower viral load in animals treated with molnupiravir or nirmatrelvir either individually or in combination compared to untreated mice. However, nirmatrelvir treatment resulted in a greater decrease in viral load compared to molnupiravir. The molnupiravir/nirmatrelvir combination was also more effective at decreasing viral load than either drug alone, but this was only statistically significant in the case of molnupiravir vs the drug combination.

## Treatment with molnupiravir or nirmatrelvir or both in combination results in marked reduction of pulmonary infection and inhibits viral spread to the brain

The lung, nose, and brain of all animals were examined for any histopathological changes and the expression of viral antigen by immunohistology to determine whether treatment of the animals with molnupiravir and/or nirmatrelvir influenced the outcome of infection. The lungs of vehicle-treated, immunocompetent animals showed the typical changes previously reported in K18-hACE2 mice infected with this virus strain (26), i.e., multifocal areas with pneumocyte degeneration, type II pneumocyte activation, mild neutrophil infiltration, and mild vasculitis, with a diffuse increase in interstitial cellularity and widespread SARS-CoV-2 antigen expression in alveolar epithelial cells (Fig. 5A). In mice that had received cyclophosphamide alone, the changes were very similar,

but slightly less widespread, with some unaltered parenchyma and less extensive viral antigen expression (Fig. 5B). With molnupiravir treatment, both inflammatory processes and viral antigen expression were markedly decreased; indeed, SARS-CoV-2 antigen was only found in disseminated patches of alveoli with positive pneumocytes (Fig. 5C). With cyclophosphamide and molnupiravir treatment, the lung parenchyma was widely unaltered, and there were only small patches of inflammation and alveoli with viral antigen expression, respectively (Fig. 5D). These were further reduced in number and size in animals that had received cyclophosphamide and nirmatrelvir (Fig. 5E). Treatment with all three compounds, cyclophosphamide, molnupiravir, and nirmatrelvir, resulted in widely unaltered lung parenchyma with no or minimal viral antigen expression (Fig. 5F).

Examination of the heads using longitudinal sections (midline) revealed consistent and widespread infection of the brain in animals treated with the vehicle or with cyclophosphamide alone (Fig. 6A and B); this was associated with mild perivascular mononuclear infiltration, in particular, in the brain stem, as described before in K18-hACE2 mice infected with this virus strain (37). In both groups of animals, immunohistology confirmed viral antigen expression in the respiratory and/or olfactory epithelium, in the latter with evidence of infection in olfactory sensory neurons (Fig. 6A and B). In the other groups, there was no evidence of viral infection of the brain (Fig. 6C through F), and viral antigen expression in the nasal mucosa was not seen or restricted to scattered individual epithelial cells. In vehicle control and cyclophosphamide mice, the nasal mucosa harbored viral antigen at this stage, in the respiratory epithelium and in the olfactory epithelium; in the latter, it also appeared to be present in sensory neurons. Consequently, the virus had reached and spread widely in the brain where it was detected in neurons; the infection was associated with mild inflammatory response, in particular, in the brain stem, as described before in K18-hACE2 mice infected with this virus strain (26, 37). After treatment with all three compounds, cyclophosphamide, molnupiravir, and nirmatrelvir, the lung parenchyma was basically unaltered, with no or minimal viral antigen expression. In all groups of mice, viral antigen expression in the nasal mucosa was not seen or restricted to scattered individual epithelial cells, and there was no evidence of viral infection of the brain, suggesting that the antiviral treatment blocked infection of the brain. Whether the latter is purely a consequence of reduced viral replication in the upper respiratory tract cannot be assessed in the present study; it does, however, appear likely.

## Evaluation of dominant and minor variants in SARS-CoV-2

To determine the impact of immunosuppression on viral diversity, 116 RNA samples from swabs and tissue were sequenced and analyzed using the EasySeq WGS protocol

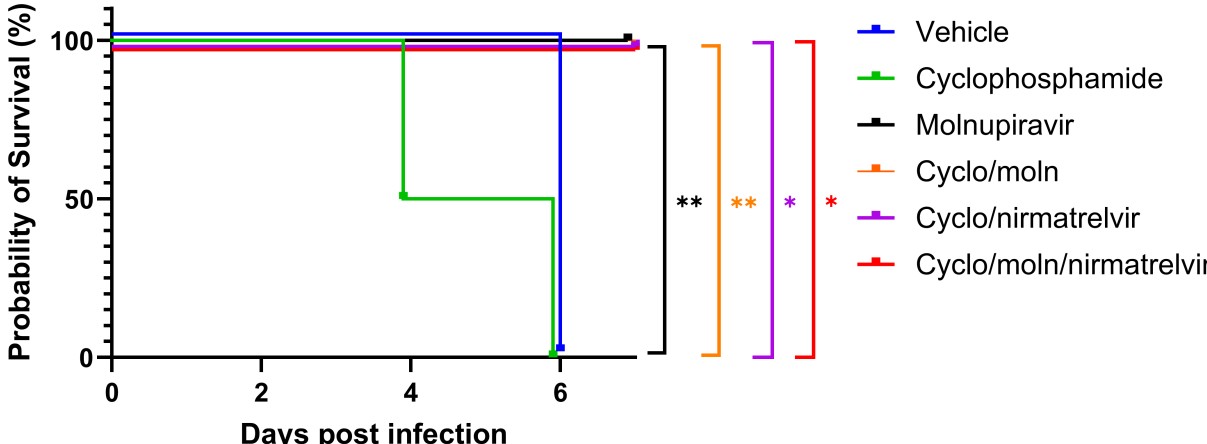

**FIG 3** Treatment of SARS-CoV-2-infected mice leads to enhanced survival. K18-hACE2 mice were challenged intranasally with $10^4$ PFU SARS-CoV-2. Survival was assessed at indicated time points, and significance was determined using the log-rank (Mantel-Cox) test ($n = 4$).

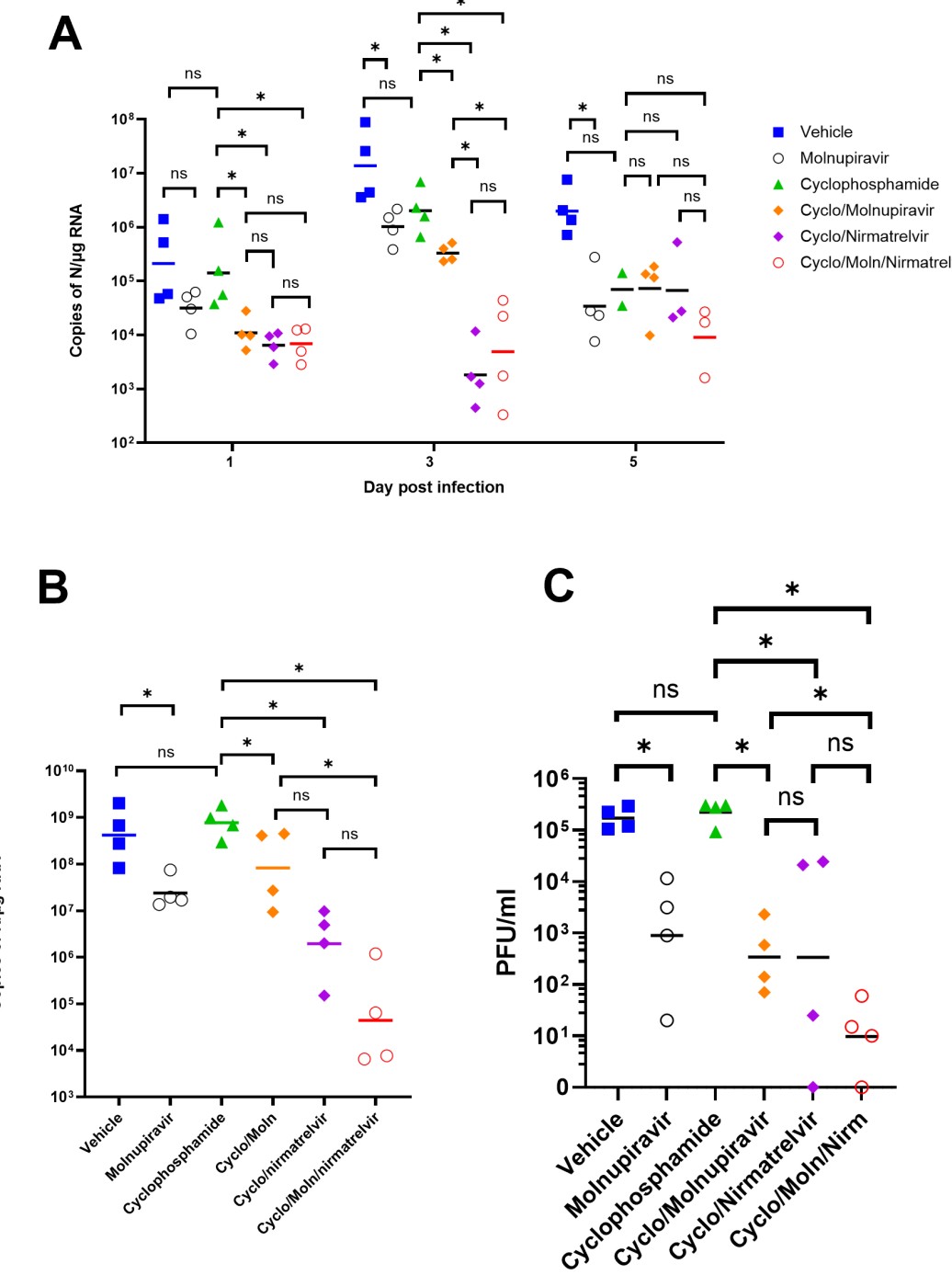

**FIG 4** Viral loads in swabs and tissues. K18-hACE2 mice were challenged intranasally with $10^4$ PFU SARS-CoV-2 and treated as indicated ($n = 4$ per group). RNA extracted from oral/throat swabs and nasal tissue was analyzed for virus RNA load using qRT-PCR and primers specific for the SARS-CoV-2 N gene. Assays were normalized relative to levels of 18S RNA. Lung tissue was analyzed for live virus by plaque assay. Data for individual animals are shown with the median value represented by a black line. (A) Throat swabs, (B) nasal tissue, and (C) lung tissue. Comparisons were made using two-way ANOVA (Bonferroni post-test) in panel A and Mann-Whitney U-test (panels B and C). * represents $P < 0.05$.

by Nimagen. Alignment files and associated index files were inputted into DiversiTools to provide mutation data, and outputs were analyzed in R. Samples with less than 90%

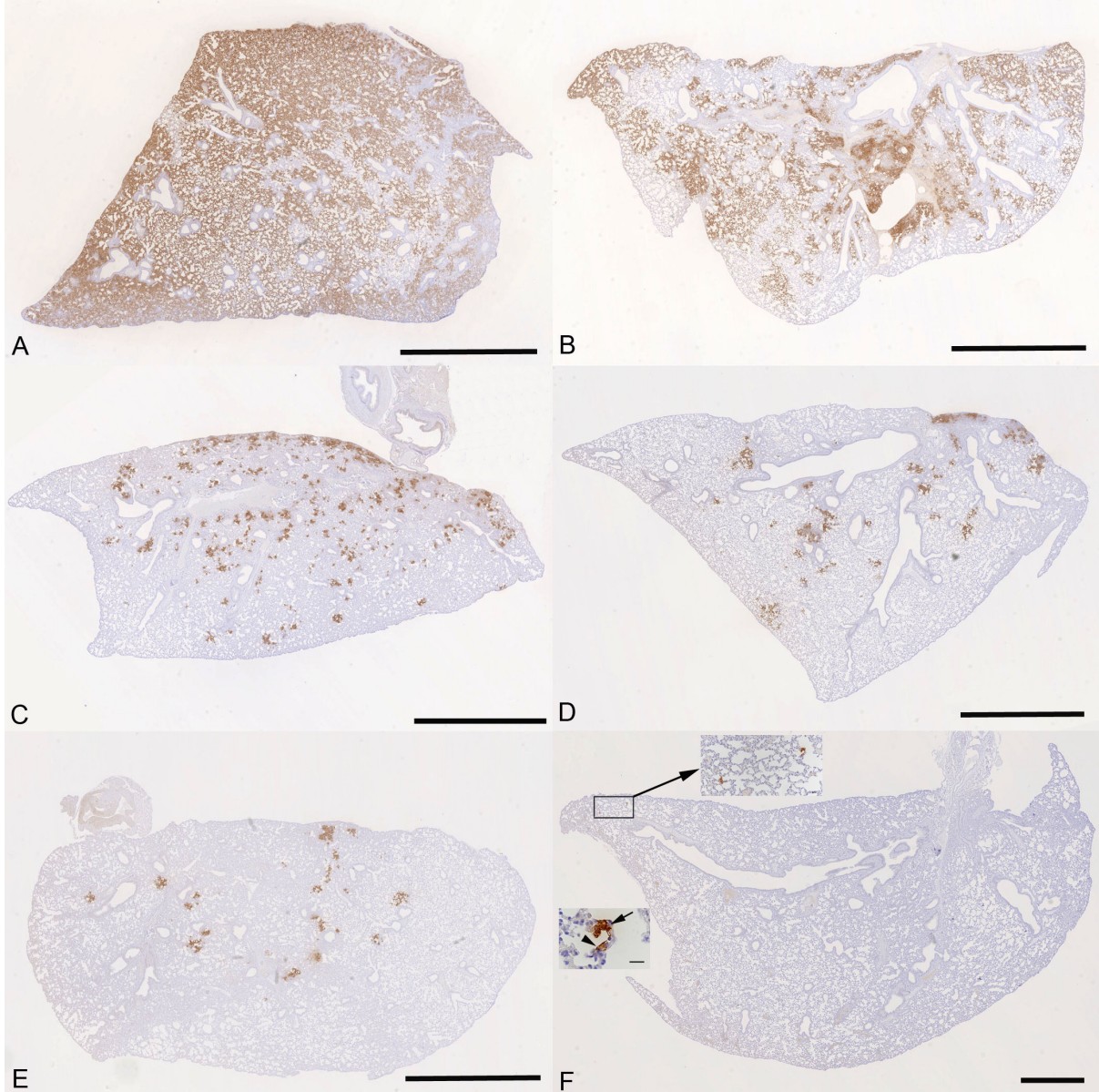

**FIG 5** K18-hACE2 mice were challenged intranasally with $10^4$ PFU SARS-CoV-2 and treated as indicated below ($n = 4$ per group). Immunohistology for the detection of viral antigen in the lung at day 6 or 7 post-infection. Sections from the formalin-fixed, paraffin-embedded left lung lobe were stained using anti-SARS-CoV nucleoprotein and counterstained with hematoxylin. Representative images from the individual treatment groups are shown as follows: (A) vehicle; (B) cyclophosphamide; (C) molnupiravir; (D) cyclophosphamide and molnupiravir; (E) cyclophosphamide and nirmatrelvir; (F) cyclophosphamide, molnupiravir, and nirmatrelvir. Viral antigen expression is restricted to pneumocytes in a few individual alveoli (higher magnifications in insets). Bars represent 2.5 mm (A–E), 1 mm (F), and 20 µm (F, insets).

breadth of coverage were discarded for mutational analysis ($n = 12$), as well as samples that returned bad or mediocre quality scores in Nextclade ($n = 13$). The samples that were excluded were associated with higher Ct values and later time points belonging in the nirmatrelvir treatment groups. Sequencing data from 89 samples were taken forward in the analysis (swab, $n = 50$, tissue $n = 39$, Table S1).

The input virus contained five substitutions and three amino acid substitutions in comparison to the reference sequence (NC_045512.2) and were thus not considered as changes during the analysis (Table 1). The S:H655Y mutation was present in 76% of the genomes that passed quality contol (QC) at the dominant level and observed as a minor

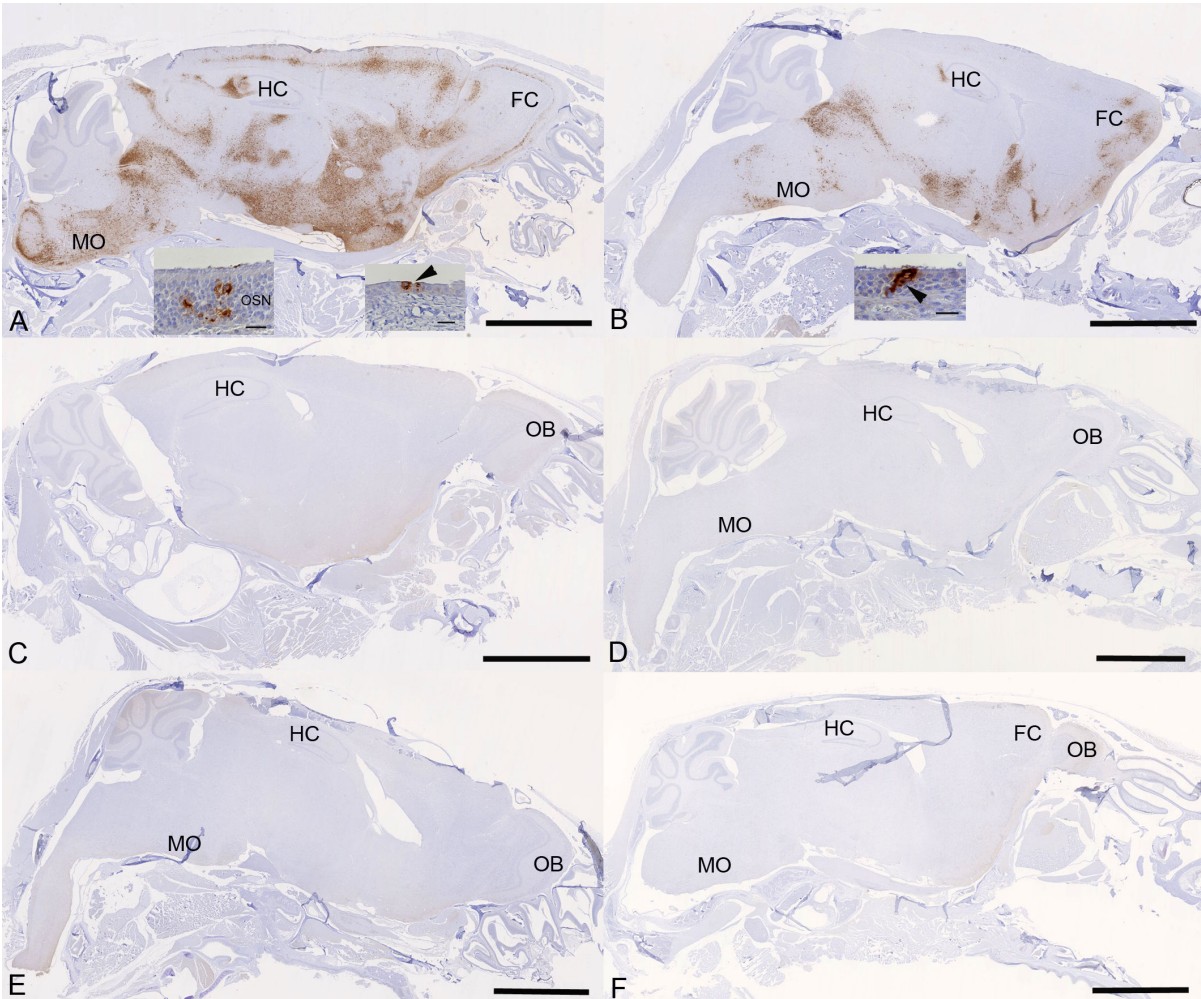

**FIG 6** K18-hACE2 mice were challenged intranasally with $10^4$ PFU SARS-CoV-2 and treated as indicated below ($n$ = 4 per group). Immunohistology for the detection of viral antigen in the brain and nose at day 6 or 7 post-infection. Sections from formalin-fixed, decalcified, and paraffin-embedded heads after longitudinal sawing in the midline were stained using anti-SARS-CoV nucleoprotein and counterstained with hematoxylin. Only small fragments of nasal mucosa were available for the examination, as the nasal turbinates had been sampled for PCR. Representative images from the individual treatment groups are shown as follows: (A) vehicle. There is widespread infection of the brain. The insets show infection of individual cells with the morphology of olfactory sensory neurons and epithelial cells in the olfactory epithelial layer (left inset) and individual respiratory epithelial cells in the nasal mucosa (arrowhead; right inset). (B) Cyclophosphamide. There is widespread infection of the brain. The inset shows a group of positive epithelial cells/sensory neurons in the olfactory epithelial layer (arrowhead). (C) Molnupiravir. There is no evidence of brain infection. (D) Cyclophosphamide and molnupiravir. There is no evidence of brain infection. (E) Cyclophosphamide and nirmatrelvir. There is no evidence of brain infection. (F) Cyclophosphamide, molnupiravir, and nirmatrelvir. There is no evidence of brain infection. Bars represent 2.5 mm (A–F) and 20 µm (A, B insets). FC, frontal cortex; HC, hippocampus; MO, medulla oblongata; OB, olfactory bulb; OSN, olfactory sensory neurons.

variant across all samples (Fig. S2). This mutation has been reported previously as a spike adaptation to other species such as cats, hamsters, and mink (54–56) and, of course, has independently arisen in human lineages such as Omicron (57). As this mutation was clearly associated with a species adaptation, it was disregarded for the evaluation of treatment and immune status-driven mutations. The other mutations appear to be novel at the time of writing; however, no distinct group was associated with driving these mutations and can be overall interpreted as a rare event (Table S2). The sequences showing the highest number of mutations were sequences derived from tissue samples. The species-specific adaptation S:H655Y was observed more frequently in the data set, where there was little evidence of adaptations specific to immunocompromised and antiviral environments, putting the evolutionary pressures into perspective. Adaptations

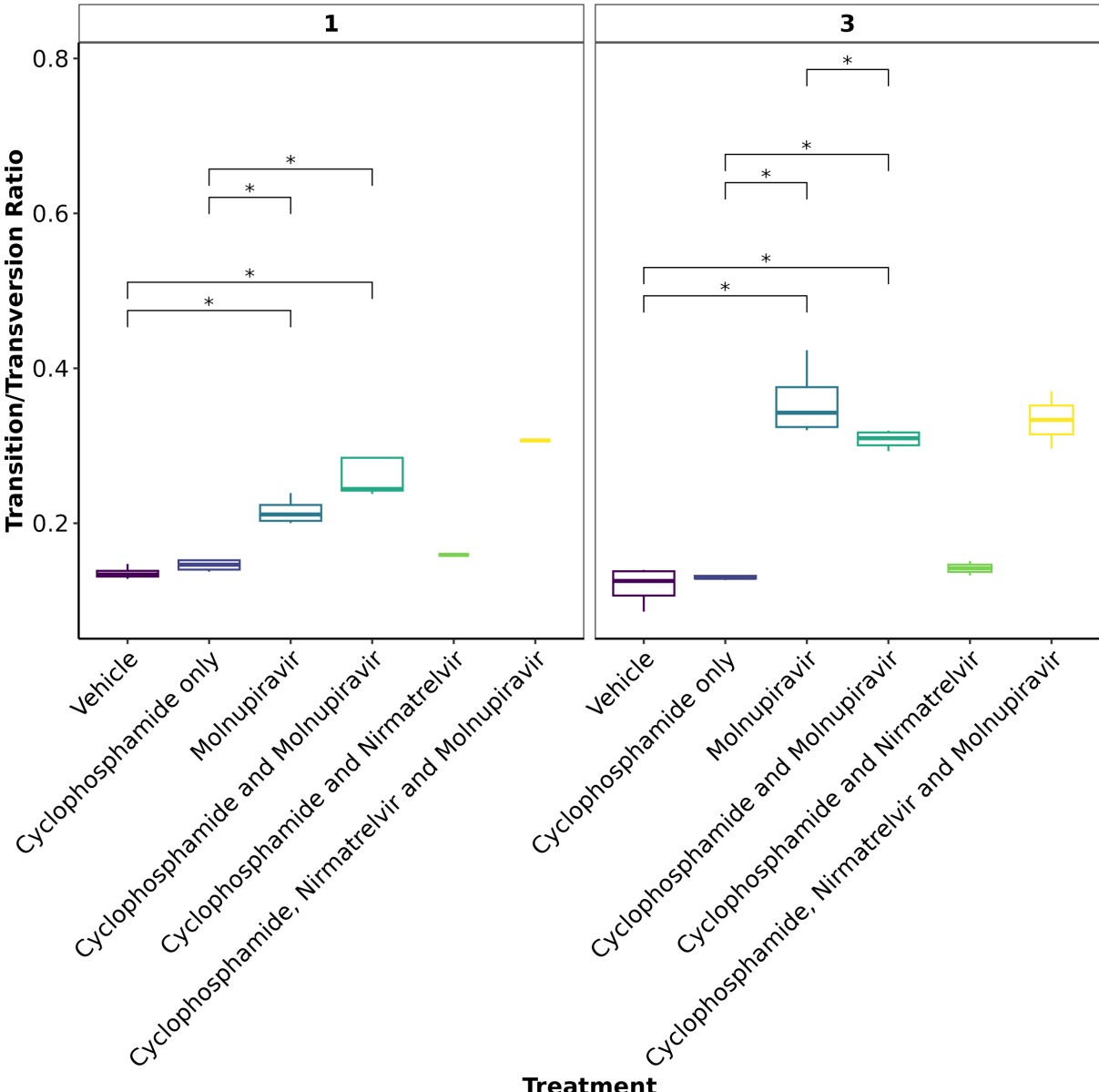

**FIG 7** The mean Ts/Tv ratio per genome plotted as boxplots. The plot is faceted by day post-infection. Fewer genomes were recovered for cyclophosphamide and nirmatrelvir and cyclophosphamide, nirmatrelvir, and molnupiravir; therefore, statistical analysis returns the differences as non-significant. Trends can be concluded with caution. * represents a *P*-value <0.05 (Mann-Whitney U-test).

associated with immunosuppression and antiviral treatment may emerge in the context of persistent replication of the virus, which would need to be investigated.

## Molnupiravir increases the Ts/Tv ratio at the minor variant level in genomes derived from swabs

To further assess the impact of immunocompromising mice by cyclophosphamide and the therapeutic agents molnupiravir and nirmatrelvir, a minor variant analysis was conducted on samples derived from throat swabs as performed previously (22). Only samples with a >90× coverage with a 100× depth were taken forward into this analysis, and the average depth was over 1,400 for each sample (Fig. S3). No relationship was observed between Ct value and the calculated average Ts/Tv (Fig. S4). The average Ts/Tv ratio for SARS-CoV-2 genomes from each mouse and the mean of each group

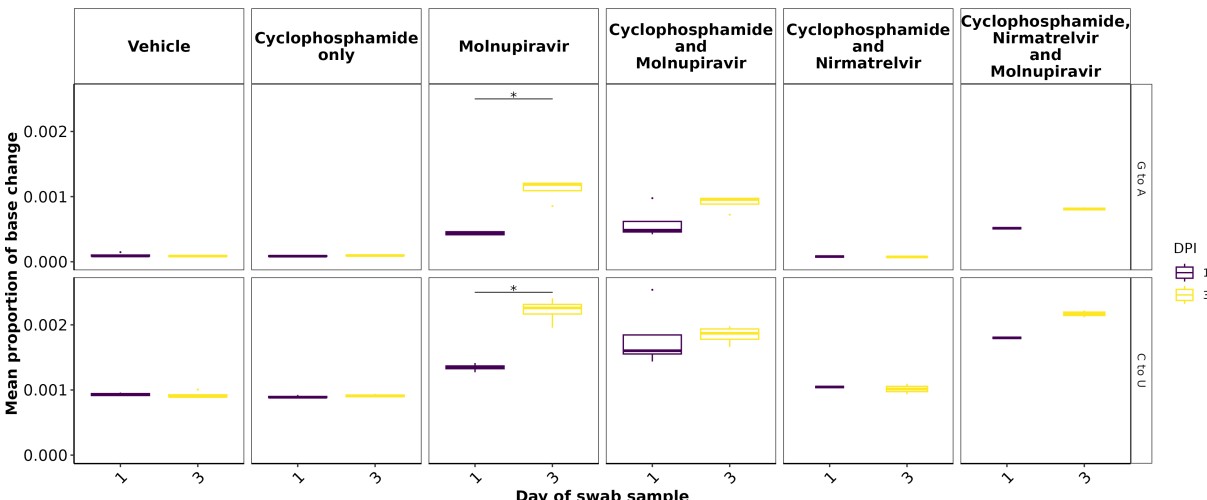

**FIG 8** C to U and G to A minor variation changes significantly increased between day 1 and day 3 post-infection in the molnupiravir-only group. A similar trend is observed between other groups, including molnupiravir treatment; however, the change is not reported as significant. * represents a *P*-value <0.05 (Mann-Whitney U-test).

were compared across cohorts at day 1, day 3, and day 5 of infection in line with analysis performed previously in a phase II clinical trial (22). On day 1, an increase in the Ts/Tv ratio was observed in the molnupiravir cohort and the cyclophosphamide and molnupiravir cohort and had a *P*-value <0.05 when compared to the vehicle control and cyclophosphamide-only groups (Fig. 7). The number of samples analyzed for cyclophosphamide and nirmatrelvir only was too small for statistical analysis; however, the trend resembles that of vehicle and cyclophosphamide only, with little change in the Ts/Tv ratio. Likewise, the combined cyclophosphamide and molnupiravir and nirmatrelvir cohort is represented by one genome derived from one mouse, due to low sequencing coverage obtained from the other mice within this group. However, the trend resembles that of other genomes with exposure to molnupiravir with an increased Ts/Tv ratio. The same is observed at day 3 of sampling; however, there is a significant difference between the mean Ts/Tv ratio between the molnupiravir-only and cyclophosphamide and molnupiravir groups. Importantly, the Ts/Tv ratios between the vehicle control and cyclophosphamide-only groups resemble each other, demonstrating that immunosuppression itself does not drive diversification of the viral genome over this time course. Curiously, when looking at base changes independently of the Ts/Tv ratios, there are significant changes between C to G, C to A, and A to U in cyclophosphamide groups on day 1 and day 3 (Fig. S3). This could be evidence of RNA editing through reactive oxygen species (ROS). Cyclophosphamide has been shown to activate oxidative stress pathways previously and may be a consequence of this treatment (58, 59). This highlights a research gap in events that can influence RNA editing in RNA viruses. The proportion of base changes in the C to U and G to A transitions was significantly different in the molnupiravir-only group, as previously seen in a phase II clinical trial (22, 23) (Fig. 8).

Further investigations are warranted to understand completely the role of immunocompromised individuals in the development of SARS-CoV-2 variants. It is more likely that immunodeficiency promotes viral persistence, providing the virus more opportunity to replicate and introduce mutations. Molnupiravir, compared to nirmatrelvir, shows a stronger mutagenic effect in this model at the minor variant level; however, data are insufficient to make conclusions regarding consensus-level changes over the time frames used in this study. When these therapies are used individually or in combination, there is successful depletion in viral load, and animals recover from infection while preventing infiltration into brain tissue. Given the concern of molnupiravir-associated lineages in circulation (23), combination therapy may reduce this through more effective

clearance of the virus (20), although this would need to be evaluated over time in a real-world setting as the mutational signatures were observed in the combined therapy group. The AGILE clinical trial is currently ongoing to answer this question (ISRCTN: ISRCTN27106947). It is important to note that the mutational spectrum reported in this study is obtained by amplicon sequencing data, and there is potential for RT-PCR errors within the data.

## ACKNOWLEDGMENTS

The authors are grateful to the technical staff at the Histology Laboratory, Institute of Veterinary Pathology, Vetsuisse Faculty, University of Zurich, for excellent technical support. J.A.H. is a member of the ISARIC4C consortium (https://isaric4c.net/about/authors/), and we thank them for the use of the SARS-CoV-2 isolate used in this study.

This work was funded by the MRC (MR/W005611/1) "G2P-UK: A national virology consortium to address phenotypic consequences of SARS-CoV-2 genomic variation" and (MR/Y004205/1) "The G2P2 virology consortium: keeping pace with SARS-CoV-2 variants, providing evidence to vaccine policy, and building agility for the next pandemic" (co-Is J.P.S. and J.A.H.) and funded in part by the US Food and Drug Administration Medical Countermeasures Initiative contract (75F40120C00085) to J.A.H. The article reflects the views of the authors and does not represent the views or policies of the FDA. Additionally, funding was aquired through DNDi under the support by the Wellcome Trust (grant ref: 222489/Z/21/Z to A.O. and J.P.S.) through the "COVID-19 Therapeutics Accelerator." A.O. acknowledges funding by Wellcome Trust (222489/Z/21/Z), EPSRC (EP/R024804/1; EP/S012265/1), and National Institute of Health (NIH) (R01AI134091; R24AI118397). J.P.S. also acknowledges funding from the Medical Research Council (MRC) (MR/R010145/1, MR/W021641/1, PA6162_G2P2-2023), BBSRC (BB/R00904X/1; BB/R018863/1; BB/N022505/1), and Innovate UK (TS/W022648/1). A.K. received support from the Swiss National Science Foundation (SNSF; IZSEZ0 213289).

## AUTHOR AFFILIATIONS

[1]Department of Infection Biology and Microbiomes, University of Liverpool, Liverpool, England, United Kingdom
[2]NIHR Health Protection Research Unit in Emerging and Zoonotic Infections, Liverpool, England, United Kingdom
[3]Laboratory for Animal Model Pathology, Institute of Veterinary Pathology, Vetsuisse Faculty, University of Zürich, Zürich, Switzerland
[4]Department of Pharmacology and Therapeutics, University of Liverpool, Liverpool, England, United Kingdom
[5]Centre of Excellence in Long-acting Therapeutics (CELT), University of Liverpool, Liverpool, England, United Kingdom
[6]A*STAR Infectious Diseases Laboratories (A*STAR ID Labs), Agency for Science, Technology and Research (A*STAR), Singapore

## AUTHOR ORCIDs

Rebekah Penrice-Randal http://orcid.org/0000-0002-0653-2097
Anja Kipar http://orcid.org/0000-0001-7289-3459
Julian A. Hiscox http://orcid.org/0000-0002-6582-0275

## FUNDING

| Funder | Grant(s) | Author(s) |
| --- | --- | --- |
| Medical Research Council | MR/W005611/1, MR/Y004205/1, MR/R010145/1, MR/W021641/1, PA6162_G2P2-2023 | Julian A. Hiscox / James P. Stewart |
| U.S. Food and Drug Administration | 75F40120C00085 | Julian A. Hiscox |

| Funder | Grant(s) | Author(s) |
| --- | --- | --- |
| Wellcome Trust | 222489/Z/21/Z | Andrew Owen |
| | | James P. Stewart |
| Wellcome EPSRC Centre for Medical Engineering | EP/R024804/1, EP/S012265/1 | Andrew Owen |
| National Institutes of Health | R01AI134091, R24AI118397 | Andrew Owen |
| Biotechnology and Biological Sciences Research Council | BB/R00904X/1, BB/R018863/1, BB/N022505/1 | James P. Stewart |
| Innovate UK | TS/W022648/1 | James P. Stewart |
| Schweizerischer Nationalfonds zur Förderung der Wissenschaftlichen Forschung | IZSEZ0 213289 | Anja Kipar |

## AUTHOR CONTRIBUTIONS

Rebekah Penrice-Randal, Data curation, Formal analysis, Investigation, Methodology, Software, Validation, Visualization, Writing – original draft, Writing – review and editing | Eleanor G. Bentley, Data curation, Formal analysis, Investigation, Methodology, Validation, Visualization, Writing – original draft, Writing – review and editing | I'ah Donovan-Banfield, Methodology, Resources, Software, Writing – review and editing | Anja Kipar, Conceptualization, Data curation, Formal analysis, Investigation, Methodology, Supervision, Validation, Visualization, Writing – original draft, Writing – review and editing | Daniele F. Mega, Data curation, Formal analysis, Investigation, Methodology, Validation, Writing – review and editing | Chloe Bramwell, Investigation, Methodology, Writing – review and editing | Joanne Sharp, Data curation, Investigation, Methodology, Project administration, Writing – review and editing | Andrew Owen, Conceptualization, Funding acquisition, Supervision, Writing – original draft, Writing – review and editing | Julian A. Hiscox, Conceptualization, Funding acquisition, Writing – original draft, Writing – review and editing | James P. Stewart, Conceptualization, Funding acquisition, Investigation, Supervision, Visualization, Writing – original draft, Writing – review and editing.

## ETHICS APPROVAL

Animal work was approved by the local University of Liverpool Animal Welfare and Ethical Review Body and performed under UK Home Office Project LicenceLicense PP4715265.

## ADDITIONAL FILES

The following material is available online.

### Supplemental Material

**Supplemental data (Spectrum01829-24-s0001.xlsx).** SRA information.
**Supplemental material (Spectrum01829-24-s0002.docx).** Tables S1 and S2; Fig. S1 to S5.

### Open Peer Review

**PEER REVIEW HISTORY (review-history.pdf).** An accounting of the reviewer comments and feedback.

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
