## [Reviewer comments · Microbiology Spectrum]

Microbiology Spectrum

The effect of molnupiravir and nirmatrelvir on SARS-CoV-2 genome diversity in severe models of COVID-19.

Rebekah Penrice-Randal, Eleanor Bentley, Parul Sharma, Adam Kirby, I'ah Donovan-Banfield, Anja Kipar, Daniele Mega, Chole Bramwell, Joanne Sharp, Andrew Owen, Julian Hiscox, and James Stewart

Corresponding Author(s): Rebekah Penrice-Randal, University of Liverpool

Review Timeline:

Submission Date:	August 1, 2024
Editorial Decision:	November 6, 2024
Revision Received:	January 14, 2025
Editorial Decision:	March 3, 2025
Revision Received:	March 4, 2025
Accepted:	March 10, 2025

Editor: Tao Deng

Reviewer(s): Disclosure of reviewer identity is with reference to reviewer comments included in decision letter(s). The following individuals involved in review of your submission have agreed to reveal their identity: Kaan Çeylan (Reviewer #2)

Transaction Report:

DOI: <https://doi.org/10.1128/spectrum.01829-24>

Re: Spectrum01829-24 (The effect of molnupiravir and nirmatrelvir on SARS-CoV-2 genome diversity in severe models of COVID-19.)

Dear Dr. Rebekah Penrice-Randal:

Thank you for the privilege of reviewing your work. Below you will find my comments, instructions from the Spectrum editorial office, and the reviewer comments.

Revision Guidelines

Sincerely,
Tao Deng
Editor
Microbiology Spectrum

Reviewer #1 (Comments for the Author):

Overall, quite well written and very interesting piece of work. Molnupiravir causing a high proportion of G-to-A and C-to-T mutations is well known from various works - including citation 23 (Sanderson et al) referenced here, which does limit the novelty of things, although most work was previously done a consensus level data. The low freq variant analysis will need to be improved, or more details on the quality filters being used if currently applied.

I was a little confused as to what is being reported here. The abstract says A>G rather than G>A - is that the case?

Intro 58 - SARS-CoV-2 abbreviation needs to be defined. Perhaps give the relevant taxonomic information for the virus as well.

The bioinformatics workflow used is appropriate for consensus level sequence generation. The analysis of within-host diversity requires different levels of filtering - more aggressive quality score filtering, much higher minimum coverage thresholds, strand bias tests, minimum variant frequency test to account for RT-PCR errors, etc etc. This does not seem to have been done has it? It seems that the BAM files from the consensus sequence generation have simply been run through DiversiTools - DiversiTools is a 'what you see is what you get' tool - it simply reports the number of As Cs Gs and Ts with limited tests (there are a couple if reports - but you have to use/evaluate them yourselves). What level of filtering and quality assessment of minority variants has been done in this study? How comparable are the samples in terms of coverage and quality etc. Given that one of the main output of the paper is the low freq variant analysis under different drugs - there needs to be much more detail on how the low frequency variant analysis was actually done? At a minimum I would suggest more aggressively filtering the input reads (presuming the current filtering is slightly looser for consensus generation) and minimum base quality and coverage thresholds for individual sites when calling a minority variant. For example, if you see ONE mutation at a site of coverage of a hundred are you considering it to be real? Etc If your average quality is 20 (error of one in a thousand) and you see a mutation at a frequency below that - are you considering it to be real? Etc More details need to be given on the lowfreq analysis if such things have already been considered/applied.

Results 402 - breadth of coverage is a starting point - but average depth, and individual depth at a specific site are equally important - what is the average depth of each sample and what was the minimum cov of a site to consider a mutation to be real - and what about min freq? Are the number of input viral genomes into the PCR normalised across samples - as if a sample has a lower copy number, RT-PCT errors will have higher frequency in the data.

Infection Virus

There is limited detail on the starting virus. NO - Having gone through the whole paper - the information is there, but confusingly spread out.

The abstract says that mice were infected with "a B lineage of SARS-CoV-2." Whereas the methods (line 126) give the brief "as previously described". There is a table of mutations in SupplInfo 2 - this table needs to be here in methods. Is this really a simple 'B' lineage - with 5 mutations and 3 AA subs - I would think it was a B daughter - can you clarify the exact lineage? One of your early papers - where this virus originates - says that the sample was ARTIC sequenced - so data is available - has the same Transition/Transversion analysis been done on this original sample?

Transition/Transversion - C to U and G to A are presented in Figure 8 (I can't seem to see the legend for this Figure?) -I think a comparison of all the base to base mutations is needed (perhaps in SupplInfo) similar to this Figure - it currently feels like a summary level without providing the underlying data

It seems the original virus lacks even D614G - do you expect the results to change at all with a more 'modern' variant of the virus?

Line 452 - previous work - I think paper number 23 should also be mentioned here

Reviewer #2 (Comments for the Author):

Dear author,

The review of the study titled The effect of molnupiravir and nirmatrelvir on SARS-CoV-2 genome diversity in severe models of COVID-19 has been completed. It would be appropriate to review the points mentioned below.

1. How the sample size was determined should be explained in more detail.
2. Although the findings of the study are clearly presented at the end of the text, the findings are not discussed sufficiently.
3. English editing is required.

I wish good work.

Dear Editor,

We would like to thank you and the peer reviewers for reading and reviewing our research article. The review from reviewer one addresses only one section of the manuscript, and we believe the changes made from this peer review has improved the article. Given that neither of the reviewers have highlighted the other sections within the manuscript such as the animal study, qPCRs and pathology, that these adjustments will be sufficient for consideration of publishing. We have responded to the reviewers below, where we have highlighted the changes made or made appropriate rebuttals.

Thanks again for your consideration.

Dr Penrice-Randal.

Response to reviewers

Reviewer #1 (Comments for the Author):

Overall, quite well written and very interesting piece of work. Molnupiravir causing a high proportion of G-to-A and C-to-T mutations is well known from various works - including citation 23 (Sanderson et al) referenced here, which does limit the novelty of things, although most work was previously done a consensus level data. The low freq variant analysis will need to be improved, or more details on the quality filters being used if currently applied.

Response: We thank the reviewer for their comments and feedback on our work and are in agreement with the improved minor variant/low frequency analysis. We used the same pipeline that was accepted in Nature Communications, and therefore feel the methodology is suffice for what we are saying in this manuscript. We agree with the points made, but do not feel it is necessary for this analysis. We have provided more information and discussion to highlight the limitations to the reader. The key message of this manuscript is to highlight that these signatures are observed in combination with nirmatrelvir, which is now being taken to clinical trial: ISRCTN27106947, therefore, we believe there is merit in this publication.

I was a little confused as to what is being reported here. There abstract says A>G rather than G>A - is that the case?

Response: We thank the reviewer for spotting this error. This has been corrected.

Intro 58 - SARS-CoV-2 abbreviation needs to be defined. Perhaps give the relevant taxonomic information for the virus as well.

Response: Thank you for this observation. We have now defined the abbreviation 'SARS-CoV-2' in the introduction as 'Severe Acute Respiratory Syndrome Coronavirus 2'

The bioinformatics workflow used is appropriate for consensus level sequence generation. The analysis of within-host diversity requires different levels of filtering - more aggressive quality score filtering, much higher minimum coverage thresholds, strand bias tests, minimum variant frequency test to account for RT-PCR errors, etc etc. This does not seem to have been done has it? It seems that the BAM files from the consensus sequence generation have simply been run through DiversiTools - DiversiTools is a 'what you see is what you get' tool - it simply reports the number of As Cs Gs and Ts with limited tests (there are a couple if reports - but you have to use/evaluate them yourselves). What level of filtering and quality assessment of minority variants has been done in this study? How comparable are the samples in terms of coverage and quality etc. Given that one of the main output of the paper is the low freq variant analysis under different drugs - there needs to be much more detail on how the low frequency variant analysis was actually done? At a minimum I would suggest more aggressively filtering the input reads (presuming the current filtering is slightly looser for consensus generation) and minimum base quality and coverage thresholds for individual sites when calling a minority variant. For example, if you see ONE mutation at a site of coverage of a hundred are you considering it to be real? Etc If your average quality is 20 (error of one in a thousand) and you see a mutation at a frequency below that - are you considering it to be real? Etc More details need to be given on the lowfreq analysis if such things have already been considered/applied.

Response:

We appreciate the reviewer's focus on the details of low-frequency variant analysis. While our aim is to identify broader global changes rather than specific iSNVs, our methodology adheres to the standards of our previously published work in Nature Communications. DiversiTools, which we employed here, functions by counting base occurrences, and its outputs include the AvQual score, providing base-calling accuracy. We have applied a minimum coverage threshold of 100X, ensuring a base-calling error rate below 2×10^{-6} . This approach is appropriate for our focus on aggregate mutation patterns. If the study aimed to identify specific iSNVs, more aggressive filtering or a different methodology would indeed be necessary. However, as the primary goal is to observe global mutational signatures, we believe this method is robust and consistent with our prior studies. We have added details in the manuscript to clarify these points and discuss potential limitations for transparency.

Results 402 - breadth of coverage is a starting point - but average depth, and individual depth at a specific site are equally important - what is the average depth of each sample and what was the minimum cov of a site to consider a mutation to be real - and what about min freq? Are the number of input viral genomes into the PCR normalised across samples - as if a sample has a lower copy number, RT-PCT errors will have higher frequency in the data.

Response: total RNA was normalised to 20ng/ul before qPCR and sequencing prep, which is now highlighted in the material and methods. The samples were arranged by

ct value during sequencing prep and the amount of PCR products added into the sequencing libraries was dependent on the initial ct value. We have added in a supplementary figure to show that there is no relationship between ct value and the mean Ts/Tv ratio that we have used in this study.

Infection Virus

There is limited detail on the starting virus. NO - Having gone through the whole paper - the information is there, but confusingly spread out.

Response:

Thanks to the reviewer for highlighting this. The supplementary table outlining the mutations in the input virus have been moved to the methods section.

The abstract says that mice were infected with "a B lineage of SARS-CoV-2." Whereas the methods (line 126) give the brief "as previously described". There is a table of mutations in SuppInfo 2 - this table needs to be here in methods. Is this really a simple 'B' lineage - with 5 mutations and 3 AA subs - I would think it was a B daughter - can you clarify the exact lineage? One of your early papers - where this virus originates - says that the sample was ARTIC sequenced - so data is available - has the same Transition/Transversion analysis been done on this original sample?

Response:

The table has been moved to methods. The stock virus has not been sequenced with the same methodology, e.g. easySeq on the illumina platforms, therefore the transition/transversion ratios will not be comparable, and there is not scope to do this for this work. We have described this as a B daughter lineage for the reviewer.

Transition/Transversion - C to U and G to A are presented in Figure 8 (I can't seem to see the legend for this Figure?) -I think an comparison of all the base to base mutations is needed (perhaps in SuppInfo) similar to this Figure - it currently feels like a summary level without providing the underlying data

Response: We thank the reviewer for pointing this out. We have added a comprehensive comparison of base-to-base mutations, similar to Figure 8, in the supplementary materials. This should provide a more detailed view of the underlying data and complement the summary-level results presented in the main figure.

It seems the original virus lacks even D614G - do you expect the results to change at all with a more 'modern' variant of the virus?

Response: We appreciate the reviewer's concern regarding the starting virus lineage. The input virus was sequenced using ARTIC and its mutations are presented in the

revised methods section. As the stock virus was not sequenced using the same easySeq methodology applied to subsequent samples, direct comparison of transition/transversion ratios is not feasible. However, our primary focus is on the mutations arising during treatment, and previous studies have shown consistent results with molnupiravir across multiple variants of concern (Prince et al., 2023). We have clarified these points in the revised manuscript.

- <https://www.biorxiv.org/content/10.1101/2021.11.23.469695v3.abstract>

Line 452 - previous work - I think paper number 23 should also be mentioned here

Response: Citation has been added.

Reviewer #2 (Comments for the Author):

Dear author,

The review of the study titled The effect of molnupiravir and nirmatrelvir on SARS-CoV-2 genome diversity in severe models of COVID-19 has been completed. It would be appropriate to review the points mentioned below.

1. How the sample size was determined should be explained in more detail.

Response: We thank the reviewer for this comment. As mentioned in the methods section; the sample size was determined based on prior studies in this field and logistical constraints, ensuring statistical power while minimising animal use in line with ethical guidelines.

2. Although the findings of the study are clearly presented at the end of the text, the findings are not discussed sufficiently.

Response: We have carefully read the discussion again to try and find where we failed to sufficiently discuss the findings. We have amended the text whenever we felt this was necessary and hope that the reviewer would point us to the right direction should they still consider the discussion as insufficient.

3. English editing is required.

Response: Manuscript was reviewed for language and revised where we felt it was appropriate, however, as the reviewer has not specified specific language shortcomings and as the native speaker coauthors cannot find serious flaws, we feel that we cannot improve the English language further at this stage.

I wish good work.

Response: Thank you very much.

Re: Spectrum01829-24R1 (The effect of molnupiravir and nirmatrelvir on SARS-CoV-2 genome diversity in severe models of COVID-19.)

Dear Dr. Rebekah Penrice-Randal:

Thank you for the privilege of reviewing your work. The two reviewers have further modification suggestions. Below you will find these comments, please revise the m/s accordingly.

Revision Guidelines

Sincerely,
Tao Deng
Editor
Microbiology Spectrum

Reviewer #1 (Comments for the Author):

- 1) Line 135 - I don't think it is enough to say 'B daughter lineage' - the actual lineage of the sequence assigned by Pangolin needs to be provided - if it can't be added then an explanation needs to be added here in the paper as to why it can't.
- 2) Line 232 - I had suggested previously that the variant calling side of things needed some improvement for low freq variants: at a minimum: (A) You say that a minimum coverage of 100 is applied - a clarification statement should therefore be added saying (if the following is true) that even if a mutation is only observed once it will be considered true if the overall coverage at that site

is greater than 100 [as I said before I think this should be improved, but at a minimum it needs to be made clear to the user what this implies]; (B) Line 235 - "the average quality scores, derived from phred, for each position was less than 2×10^{-6} " - this is no longer a quality score, this is now the probability of error that the phred represents - but the point is I don't actually think that value is possible is it? The max phred score in Illumina is Q40 - which is an error rate of 1×10^{-4} - if every base in every read aligned at a site was Q40 you average error rate would still be 1×10^{-4} - so how can you get an error rate lower than that? Some clarification is needed.

Reviewer #2 (Comments for the Author):

Dear author,

The review of the revised version of the study titled The effect of molnupiravir and nirmatrelvir on SARS-CoV-2 genome diversity in severe models of COVID-19 has been completed. My only suggestion after the corrections is to review the spelling rules in the references and if it is not required to write the names of all the authors in the references according to the journal policies, it would be appropriate to use the expression et al. after a certain number of authors.

I wish good work.

Dear Editor,

We sincerely appreciate the time and effort you and the peer reviewers have dedicated to evaluating our manuscript. We acknowledge that Reviewer #1 has raised concerns regarding the pangolin lineage assignment, which we had previously adjusted in response to their initial comments. However, we maintain that the virus lineage is best described as B, as originally stated. We believe it is sufficient to refer to this as lineage B while highlighting the relevant mutations in Table 1.

We have further clarified the bioinformatics methodology section, which we hope resolves Reviewer #1's concerns. Additionally, we agree that the revisions have strengthened the manuscript, and we are grateful for the constructive feedback that has contributed to these improvements.

Thank you again for your consideration.

Dr Penrice-Randal.

Response to reviewers

Reviewer #1 (Comments for the Author):

1) Line 135 - I don't think it is enough to say 'B daughter lineage' - the actual lineage of the sequence assigned by Pangolin needs to be provided - if it can't be added then an explanation needs to be added here in the paper as to why it cant.

Response: We have rechecked the lineage assignment using the latest version of Pangolin, which continues to classify the virus as lineage B. In our previous revision, we adjusted the wording to "B daughter lineage" in response to the reviewer's comments; however, upon further verification, our original designation was correct. We have therefore reverted to describing the virus as a B lineage virus, with the relevant mutations detailed in Table 1. We believe this accurately reflects the data, and no further changes are necessary in response to this comment.

2) Line 232 - I had suggested previously that the variant calling side of things needed some improvement for low freq variants: at a minimum:

(A) You say that a minimum coverage of 100 is applied - a clarification statement should therefore be added saying (if the following is true) that even if a mutation is only observed once it will be considered true if the overall coverage at that site is greater than 100 [as I said before I think this should be improved, but at a minimum it needs to be made clear to the user what this implies];

Response: Thank you for this suggestion. We have clarified this aspect of the methodology to ensure transparency. Specifically, for each genome, we calculated the average transition/transversion ratio, which provides a measure of relative global changes in mutations, assuming consistent sequencing error

rates. Additionally, we have included a link to the code in the manuscript, allowing readers to examine the approach in detail.

(B) Line 235 - "the average quality scores, derived from phred, for each position was less than 2×10^{-6} " - this is no longer a quality score, this is now the probability of error that the phred represents - but the point is I don't actually think that value is possible is it? The max phred score in Illumina is Q40 - which is an error rate of 1×10^{-4} - if every base in every read aligned at a site was Q40 you average error rate would still be 1×10^{-4} - so how can you get an error rate lower than that? Some clarification is needed.

Response: Thank you for pointing this out. This statement was added in response to previous reviewer comments, but we agree that it may be unclear. The reported value is directly generated by DiversiTools as "AvQual: the average base-calling error probability, which is related to the Phred quality score." We have revised the methodology section to provide greater clarity on this process and its interpretation. We hope this resolves the concern, but we are happy to further clarify if needed.

Reviewer #2 (Comments for the Author):

Dear author,

The review of the revised version of the study titled The effect of molnupiravir and nirmatrelvir on SARS-CoV-2 genome diversity in severe models of COVID-19 has been completed. My only suggestion after the corrections is to review the spelling rules in the references and if it is not required to write the names of all the authors in the references according to the journal policies, it would be appropriate to use the expression et al. after a certain number of authors.

I wish good work.

Response: Thank you for your review and helpful feedback. We have formatted the references in accordance with the journal's requirements but are happy to make any necessary adjustments during the editing process if needed.

Re: Spectrum01829-24R2 (The effect of molnupiravir and nirmatrelvir on SARS-CoV-2 genome diversity in severe models of COVID-19.)

Dear Dr. Rebekah Penrice-Randal:

Your manuscript has been accepted, and I am forwarding it to the ASM production staff for publication. Your paper will first be checked to make sure all elements meet the technical requirements. ASM staff will contact you if anything needs to be revised before copyediting and production can begin. Otherwise, you will be notified when your proofs are ready to be viewed.

Sincerely,
Tao Deng
Editor
Microbiology Spectrum

Reviewer #1 (Comments for the Author):

Authors have addressed all comments